# Marinades Based on Natural Ingredients as a Way to Improve the Quality and Shelf Life of Meat: A Review

**DOI:** 10.3390/foods12193638

**Published:** 2023-10-01

**Authors:** Agnieszka Latoch, Ewa Czarniecka-Skubina, Małgorzata Moczkowska-Wyrwisz

**Affiliations:** 1Department of Animal Food Technology, University of Life Sciences in Lublin, 8 Skromna St., 20-704 Lublin, Poland; agnieszka.latoch@up.lublin.pl; 2Department of Food Gastronomy and Food Hygiene, Institute of Human Nutrition Sciences, Warsaw University of Life Sciences (WULS), 166 Nowoursynowska St., 02-787 Warsaw, Poland; malgorzata_moczkowska@sggw.edu.pl

**Keywords:** meat, pork, beef, poultry, marinating, sensory quality, microbiological quality

## Abstract

Marinating is a traditional method of improving the quality of meat, but it has been modified in response to consumer demand for “clean label” products. The aim of this review is to present scientific literature on the natural ingredients contained in marinades, the parameters of the marinating process, and certain mechanisms that bring about changes in meat. A review was carried out of publications from 2000 to 2023 available in Web of Science on the natural ingredients of meat marinades: fruit and vegetables, seasonings, fermented dairy products, wine, and beer. The review showed that natural marinades improve the sensory quality of meat and its culinary properties; they also extend its shelf life. They affect the safety of meat products by limiting the oxidation of fats and proteins. They also reduce biogenic amines and the formation of heterocyclic aromatic amines (HAAs) and polycyclic aromatic hydrocarbons (PAHs). This is possible due to the presence of biologically active substances and competitive microflora from dairy products. However, some marinades, especially those that are acidic, cause a slightly acidic flavour and an unfavourable colour change. Natural compounds in the ingredients of marinades are accepted by consumers. There are no results in the literature on the impact of natural marinades on the nutritional value and health-promoting potential of meat products, so it can be assumed that this is a future direction for scientific research.

## 1. Introduction

Marinating is a commonly used process that improves the sensory quality of meat and its culinary properties (e.g., its yields). It is also a preservation method [1,2,3,4]. Marination involves adding liquids infused with flavourings, spices, and functional additives to meat products. Marinades are usually a mixture of various ingredients including water/oil emulsions, organic acids, extracts, mineral salts, chemical tenderizers, aromatic vegetables, fruit juices and vinegars, lemon juice, wine, soy sauce, essential oils, fermented dairy products, herbs, and spices [5,6,7,8,9,10,11]. Marination is a nonthermal technology, and some marinade components act against microbes and autooxidation [4,5,12,13]. The marinating process is applied to various types of meat, such as pork, beef, lamb, rabbit, poultry, and wild game [11,14].

Meat marinating is carried out by various methods: dipping (soaking), tumbling, and injection into the product [1,5,15,16,17,18]. The dipping method entails immersing the meat into the marinade at a low temperature for a specific period. This method is used by meat companies of various sizes and in households. The injection method involves the use of needles to apply pressure and introduce a precise amount of marinade liquid into the meat. Tumbling involves rotating the meat in a horizontally tilted drum while the marination liquid is added [19]. At home, consumers generally use the immersion technique to marinate their meats. Commercial marinades have an alkaline pH, while acidic marinades are also used in the food industry and at home [20,21]. Alkaline marinades contain phosphates, while acid marinades are usually prepared with the addition of—or are based on—organic acids or their salts [22].

Low pH values in acidic marinades increase the tenderness of the meat by lowering the pH, which in turn leads to a weakening of the muscle structure [8,9,10,11,21,23,24,25,26,27,28]. Marinades increase the meat’s water-holding capacity (WHC), reduce cooking losses, and improve the meat’s colour and juiciness [29,30]. Studies have reported that marination improves meat’s flavour and can reduce off-flavours [2,31,32,33]. The overall quality of marinated meat products is influenced by the marination method, the ingredients in the composition of the marinade, and the marination conditions (pH, time, and temperature) [1,34]. The use of marinades has been reported to improve meat’s microbiological quality by inhibiting the growth of spoilage and pathogenic microorganisms, thereby enhancing its safety [5,7,8,9,11,21,26,27,28,35,36].

Marinating has been used in the meat industry for several decades, but the process continues to be improved through ingredient selection for marinade formulation, process control, and technological approaches to improve the quality characteristics of the final meat products, especially the most desirable one, which is tenderness. To increase meat tenderness, other strategies, such as chemical and mechanical methods, are adopted [37], but recently consumers have been demanding that different products, including meat and meat products, have clean labels and their production does not involve preservatives or synthetic components [38,39].

In this review, we provide a critical appraisal of the meat marination process, including different natural ingredients and the relationship between meat quality, its physicochemical and sensory indicators, and microbiological quality. The aim of this study is to present certain natural ingredients used to prepare marinades, the parameters of the marinating process, and the mechanisms of changes in meat through marination for the desired features of products and the extension of their shelf life.

## 2. Materials and Methods

On July 2023, the online database Web of Science Core Collection (WoSCC) was accessed with the following search string: “marinating” OR “meat quality*”. The search string was applied to the title, abstract, and author keywords fields of 708 indexed publications. After adding an additional filter, “marinade”, 298 publications were returned. Other bibliographic aspects were used, such as publication year (2000–2023) or publication language (English) (Figure 1). The search yielded 272 publications, which were used for our analysis. The exclusion criteria were paid articles, articles not available to read in full, and articles about rarely consumed meat. This review compiled research articles, including types of marinade based on natural ingredients, marinade methods, marinade mechanisms, and its effects on sensory quality, safety, and shelf life (Figure 1).

## 3. Results and Discussion

### 3.1. Effect of Traditional Marinades on the Quality of Meat

Water solutions with sodium chloride and sugars [40] and NaCl with sodium tripolyphosphate [41,42], with bicarbonate [43], or with a combination of all these ingredients [43,44] are especially important in the production of traditional marinades applied to poultry [31,41,42,43,44,45] and beef [40]. Some authors indicate a more complex marinade composition for poultry, including water (80%), NaCl (3%), a sugar mixture (dextrose, lactose, and saccharose, 9.5%), wheat flour (4%), and milk proteins (3.5%) [46], as well as phosphates and polyphosphates (60%), salt (27%), carrageenan (10%), and 3% guar gum (0, 0.05, 0.1, 0.15, and 0.2%) [47].

Marinating in a traditional way improves the tenderness and flavour of meat, as well as its juiciness. It has also been used as a tenderization method because the dispersion of ions, such as sodium and phosphate, in marinades results in a tenderizing effect due to the association of the ions with the protein. The addition of phosphate salts, particularly pyrophosphate and tripolyphosphate, increases the water-holding capacity (WHC) of meat, improves the texture and product yield, and reduces cooking losses [31,45,48]. A phosphate concentration of about 0.3% or higher is believed to act on muscle proteins by increasing the pH and ionic strength, as well as specifically by complexing protein-bound Mg and Ca, which results in increased solubilization of myosin and actin (actomyosin dissociation and depolymerization of thick and thin filaments) [49]. Although phosphates have been shown to improve meat quality, several countries have banned their use in meat production [50].

Some ingredients, such as sodium bicarbonate, have been used in marinades for pork and poultry to minimize the problem of PSE (pale, soft, and exudative) meat [44,51,52,53]. Additionally, they reduce shear force and improve meat yield [43,54,55]. The effect of bicarbonates may be due to their higher buffering capacity and ionic strength than those of phosphates [51].

The WHC of meat is minimal when the pH is close to the isoelectric point of myofibrillar proteins (about 5.2–5.3 in poultry meat) [21]. Sodium chloride plays a key role in the solubilization of myofibrillar proteins for subsequent denaturation and aggregation to improve water retention and the acceptable rigidity and elasticity of the meat gels. A sodium chloride concentration of 4.6 to 5.8% is known to produce maximum swelling of myofibrils and a high level of water absorption [56].

Marinade ingredients and their pH can influence the physicochemical properties of the meat and also reduce the level of polycyclic aromatic hydrocarbons (PAHs) in grilled products. Studies [57] have shown that a marinade with an alkaline pH based on sodium bicarbonate increases the PAH content, particularly heavy PAHs, in grilled chicken meat. In turn, the addition of juices lowers the pH and inhibits PAH formation reactions [58].

Marination is a slow process in which sufficient time is required to get the maximum output, and the incorporation of chemicals is not in line with the emerging trend of consuming natural food products. The use of chemicals as tenderizing agents improves flavour and aroma to a certain extent [59]; however, too high a concentration of these can result in a bitter, sour, and metallic taste [60,61]. 

### 3.2. Effect of Plant-Based Marinades on Quality of Meat

Various sour fruits or fruit juice, vegetables (Table 1), and fruit vinegar (Table 2) as well as enzymes derived from plants have also been used as meat marinades. 

**Fruit and vegetables marinade.** Marination using fruit and vegetable juices affects the water-holding capacity of the meat, and the cooking efficiency of the product is increased [1]. Incorporating fruit and vegetable juice into marinades can enhance the aroma, flavour, juiciness, and tenderness of the meat due to the denaturation of proteins [1,73]. Reducing the meat’s pH to below the isoelectric point can increase positive charge of the myofibrillar proteins, thus causing more repulsion forces between the thick and thin filaments, which bring about the expansion of the protein network [8,21,56,83]. 

The low pH and proteolytic enzymes from plants have the potential to improve the textural properties of the meat by affecting myofibrillar proteins [10,11,21,23,24,80,84,85,86]. Several studies [73,87,88] have reported that organic acids, e.g., lemon or citrus juice in the marinating solution, derived from fruit can play a crucial role in the tenderization of marinated meat. They contribute to the degradation and solubilization of connective tissues, resulting in a reduction in muscle fibre diameter and thickness, leading to meat fibre swelling and enhancing proteolysis by cathepsins (the optimal pH for this activity is in the range of 3.5–5.0). There is also an increased conversion of collagen to gelatin at a low pH during thermal treatment. This process weakens the electrostatic interactions between myofibrillar protein chains and connective tissues, while increasing proteolysis mediated by cathepsin [23,73,85,89]. Marinades containing proteolytic enzymes or characterized by a low pH could be used as tenderizers in jerky produced from initially tough meat, such as wild boar. Marinades based on lemon and honey and also vinegar can be used for flavouring tender meat with an intense aroma [71]. The marination of pork loin slices using a solution with a mix of extra virgin olive oil, beer, and lemon juice produces an overall improvement in the technological and sensory properties of the meat, extending the shelf life to six days [68]. Marination with fruit juice affected the water-holding properties of chicken and turkey meat samples. As for textural properties, marination affected the tenderness of turkey meat positively. Turkey breast samples marinated with black grape juice were determined to be the most tender. In addition, marination with natural fruit juices significantly inhibited lipid oxidation in chicken and turkey meats compared to an unmarinated group [75].

According to Samant et al. [90], acidic marinades may have a negative impact on the sensory characteristics of the product. Other authors [91,92] have stated that the addition of citric and acetic acids did not cause any negative changes to their products studied, apart from a slight acidic flavour and an intense odour after long marinating times in some cases. The type of marinade, especially marinades based on fruit and fruit juices, affects the colour of meat, such as chicken and turkey, and in particular increases the lightness L* and b* values, which are related, for example, to the presence of carotenoids [11,21,63,74,93,94].

Marinating meat samples with fruit juices generally decreased the pH values compared to those of the controls (unmarinated), and this reduction varied among different treatments [29]. According to other authors, the effect of fruit juice extract does not significantly affect the pH of meat. For example, pomegranate juice extract applied to chicken patties did not cause significant changes [95]. However, a slight increase in pH was observed after storage when a pomegranate fruit juice phenolic solution was used on chicken breast [96]. Other authors [29,75] have reported that the cooking losses of meat samples treated with citric acid and grapefruit juice ranged from 22.4 to 33.3%. According to Nadzirah et al. [97], the cooking loss of bromelain-treated beef was higher than that of an untreated meat sample. As lemon juice contains citric acid, this ingredient is often included within marinade solutions to improve meat WHC by lowering its pH [12].

Authors reported that mulberry polyphenol significantly decreased the TBARS values of dried minced pork slices [98], and pomegranate fruit juice could delay the TBARS values in chicken samples stored at 4 °C for 28 days [96]. These effects are a result of the antioxidant character of phenolic acids, flavanols, and anthocyanins from mulberry pulp [99]; catechins, quercetin and its derivatives (flavanols), and anthocyanidins from grape pomace [100]; and polyphenols (tannins and flavonoids) and anthocyanins from pomegranate juice [67,96,101,102]. Polyphenols can act as antioxidants or pro-oxidants, depending on their concentration and interactions with the food matrix [103].

According to Jongberg et al. [104], antioxidants break free radical chains of oxidation by the donation of hydrogen atoms from the phenolic groups, thereby preventing lipid or protein oxidation. Natural antioxidants from plants have been employed to prolong the shelf life of meat by reducing lipid oxidation [39]. 

Some studies [105] suggest that marinating ingredients can reduce foodborne pathogens after application and during storage. However, certain marinade formulations have been shown to have little to no effects in reducing the microbial load of poultry products during storage.

Citric acid and acetic acid derived from fruit are added to marinades to meet food safety and quality requirements. Weak acids exhibit antimicrobial activity mainly in their undissociated form, which penetrates the cell membrane, acidifies the cytoplasm, and increases the toxic level of the dissociated acid anion. In addition, the chelating properties of organic acids (such as citric and malic acid) can also destabilize the cell’s outer membrane [106]. Other researchers [107,108] also have shown that organic acids not only have the ability to pass the outer membrane of Gram (−) bacteria [109] but they can also act as permeabilizers, enabling other hydrophobic molecules to enter the membrane. The rate at which lemon juice or citric acid deactivates pathogens depends on the acid concentration, pH, and temperature [110]. 

In previous studies, the higher antimicrobial activity of acetic acid was compared to that of citric acid [92,106,111,112,113]. Its lower molecular weight as well as its greater hydrophobicity increase its ability to penetrate bacterial membranes at a higher rate under aerobic conditions at pH values greater than 4.5 [114]. Anaerobic rather than aerobic conditions prevail during marination, which may affect the microbial inhibitory dynamics of organic acids [115]. 

Several authors [36,67,116,117] have reported the antimicrobial effect of marinating solution components regardless of the type of marination. The combination of organic acids, ethanol, and sodium chloride can strongly inhibit several microorganisms, including pathogens such as *Salmonella*, *Listeria monocytogenes*, *Escherichia coli*, and *Staphylococcus aureus* [118,119]. In turn, *Pseudomonas* spp. counts are of great importance for the shelf life and quality of marinated products, as this microbial group dominates the spoilage population of meat, producing intense off-odours and flavours [67]. Authors observed a decrease in the population of *Pseudomonas* spp. after marinating meat in lemon juice [120], in pomegranate marinade [121], and after treatment in citric acid [67], as well as the inhibition of the multiplication of lactic bacteria associated with gaseous spoilage of modified atmosphere-packaged raw chicken meat after the use of a tomato base marinade [122]. 

The lower total mesophilic aerobic and total psychrophilic aerobic counts in marinated pork samples were due to the presence of organic acids (mainly citric acid) and phenolic compounds (mainly flavonoids, tannins, anthocyanins, and gallic acid) in fruit juice. These compounds possess antimicrobial properties and can damage the bacterial cell membrane, disrupt energy production mechanisms, and disturb the intracellular ion balance, leading to increased bacterial permeability [93,123].

Vegetable marinades are less popular for the tenderization of meat (Table 3). The antioxidant activity of compounds, such as the sulphur and flavonoids of garlic and onion and their extracts, are concentration-dependent. Garlic and onion juices have demonstrated an antioxidant effect in meat and improved the flavour of pork and broilers. In one study, all the pork marinated with garlic (3%, 6%) and onion (3%) juice had significantly higher juiciness and tenderness scores than those of the control samples. In countries where garlic and onion are commonly used, there is less resistance to having strong flavours in foods. Pork marinated with garlic juice has a higher muscle pH and lower TBARS values as well as receives higher ratings for flavour, juiciness, and tenderness compared to other treatments. The addition of garlic did not result in a strong flavour, but did produce an antioxidant effect and extended the shelf life of the pork product up to 7 days [82,124].

Wang et al. [125] stated that marinades based on green, white, and yellow teas decrease PAH levels in grilled chicken wings by about 57%, 30.7%, and 23.3%, respectively. The PAH content reduction stemmed from the phenolic compounds present in teas, which can scavenge free radicals and inhibit PAH-forming reactions.

**Enzymes from fruit and vegetables**. Plant-based exogenous proteolytic enzymes, such as papain, bromelain, and ficin, play a dominant role in meat (beef and venison meat) tenderization. These natural tenderizers are extracted from various plant origins, such as fruits and vegetables (stems and leaves) [80,126,127,128,129,130,131,132]. The final texture and appearance of meat can differ according to the type, amount of enzyme, and the way of introducing it into the muscle: dipping in a solution, pumping the enzyme solution into blood vessels, or the rehydration of freeze-dried meat. However, the dipping and pumping methods were found to be unsatisfactory because of their over-tenderizing effect [78,133]. 

Kadıoğlu et al. [24] concluded that samples marinated with pineapple fruit juice for 160 min. were recommended because the tissue fibre changed to a greater extent which resulted in a better overall quality and greater economic profits. Gokoglu et al. [134] implied that the acceptance of meats treated with vegetable proteases depended on consumers’ level of familiarity with the plants and the difference in consumer tastes in different cultures.

An endogenous tenderization mechanism is activated in animal flesh and controlled by calpain systems, and this action is temperature-dependent. There are major differences in tenderized meat quality depending upon the type of enzyme [135]. Papain is extracted from papaya latex (EC 3.4.22.2) and is one of the most common plant enzymes employed for meat tenderization due to its ability to break down both myofibrillar proteins and connective tissues [136]. Bromelain from pineapple peel or juice can extensively degrade the collagen from beef, giant catfish skin [137], and wild boar [78]. According to various authors [138,139], bromelain exhibits a more essential hydrolytic action on collagen than on myofibrillar proteins, which leads to the better tenderization of tough meat.

Increases in temperature during cooking can enhance enzyme action; e.g., papain’s optimal activity occurs at 65–80 °C [16,136]. During heating, different meat proteins denature and cause structural changes in meat, such as the destruction of cell membranes, shrinkage of meat fibres, the aggregation and gel formation of myofibrillar and sarcoplasmic protein shrinkage, and the solubilization of connective tissue. It also diminishes their water-holding capacity and therefore increases water loss [140]. Specifically at 80 °C, the hardness increases due to the denaturation of myofibrillar protein [141].

Exogenous proteases should be used under controlled conditions to achieve optimum results, and their excessive usage may deteriorate product quality. The majority of exogenous enzymes are plant-based and mainly extracted through solvents [142]. 

**Fruit vinegards**. It has been observed that studies examining the effects of vinegar-based marination on meat quality have increased in recent years [10,80,143]. Examples of vinegar-based marinades are presented in Table 2.

Vinegar is a fermented product obtained from the oxidation of ethyl alcohol by acetic acid bacteria [144].

**Table 2 foods-12-03638-t002:** Selective examples of marinades based on vinegar from fruit.

Marinade Based on Fruit Vinegar	Marinating Condition	Meat	Source
organic fruit vinegars: blackberry (pH 3.32), pomegranate (pH 3.30), rosehip (pH 3.24), and grape (pH 3.22) vinegars.	24 h at 4 °C	beef	[10]
olive oil/balsamic vinegar (5%, 10%)	3, 9, and 15 days, 2–4 °C, 1:1 (ratio of meat to marinade), vacuum	pork	[68]
apple cider vinegar (70%) apple cider vinegar and pomegranate juice (35% + 35%) + olive oil, honey, thyme	1, 3, 6, and 9 h at 4, 10, and 20 °C	chicken	[67]
balsamic vinegar marinade (pH 4.4)	24 h at 4 °C	deer, wild boar	[71]
asparagus juice + traditional balsamic vinegar	4, 24, and 48 h at 4 °C	beef	[80]
aronia vinegar, grape vinegar, hawthorn vinegar	24 h at 4 °C	chicken	[85]
black chokeberry vinegar (pH: 3.75), grape vinegar (pH: 2.95), and hawthorn vinegar (pH: 3.20)	24 h at 4 °C 1:1 (ratio of meat to marinade)	beef	[145]

Vinegars can be made from various raw materials, such as fruits (aronia, grape, hawthorn), vegetables, and cereals, and are named according to these materials. The composition of vinegar can change depending on the raw material and production process used [146,147,148]. Fruit vinegars are rich in organic acids, such as acetic, tartaric, formic, lactic, citric, and malic acids, but also contain high levels of phenolic compounds, vitamins, and minerals [149,150]. 

Many authors [75,151,152,153,154] have pointed to the rise in moisture content (the water-holding capacity) of buffalo meat, beef chunk, chicken breast, and pork loin treated with plant-derived extracts and salt solutions. The significant space between the thin and thick filaments caused by myofibrillar disintegration, the degradation of the connective tissues (the perimysium and endomysium), and the dissolution of collagen affect the level of water retention by the muscles [155], resulting in the softening of the texture of the meat, for example, for beef meat samples marinated with asparagus juice and traditional balsamic vinegar. Authors [156,157] have stated that the more moisture the meat absorbs during marination, the less it loses during roasting, which improves its sensory properties and Warner–Bratzler shear force results, including its juiciness and tenderness. 

The presence of organic acids in vinegars is thought to be responsible for the lower pH of marinated meat samples [80,85], which decreases the cooking losses of meat. In these cases, meat proteins could be affected by marination with vinegar due to the low pH values. Marination with fruit vinegars causes significant deterioration in muscle fibres and irregular muscle fibres, which makes the meat more tender [10,23,75,158]. Vinegar marinades decrease the meat’s hardness and chewiness and, as a result, reduce the thickness and fibre diameter of muscle samples, which can be seen in microstructure images. In conducted tests, the marinade using grape vinegar was the most effective, in terms of the meat’s sensory properties, while marination with aronia and hawthorn vinegars negatively affected the odour properties of the samples, but it had a positive effect on the texture of poultry [10,85,145,159].

Marinating meat in vinegar (white wine, red wine, apple cider, elderflower, and apple cider with raspberry juice) led to significant PAH content reductions in grilled meat compared to control samples of meat: about a 82% PAH reduction with elderberry vinegar and a 79% reduction with white wine vinegar [160].

### 3.3. Effect of Marinades Based on Seasonings on the Quality of Meat

Various seasonings have also been used as a marinade for meat (Table 3). Marinade solutions including “natural” ingredients (e.g., spices, herbs, essential oils extracted from flowers, fruits, roots, buds and leaves through distillation processes, etc.) are widespread in the meat industry for poultry, beef, and pork meat, due to their organoleptic, antimicrobial, and antioxidant properties [5,35,116,161,162,163]. Spices and extracts with antimicrobial and antioxidant properties are also added in marinades to add flavour and to increase the shelf life of meat products [1].

**Table 3 foods-12-03638-t003:** Selective examples of marinades based on seasoning plants.

Seasoning Used to Marinade	Marinating Condition	Meat	Source
NaCl (6%, wt/vol), food-grade sodium tripolyphosphate (3%, wt/vol), thyme (0.5%), orange oil (50:50), and water (91%)	marinated for 20 min in vacuum with 10% (wt/vol) of a prechilled (4 °C) marination solution.	chicken	[15]
extract from dried sage	vacuum or assisted by ultrasound impregnation; 4 °C; rotated at 75 rpm; 180 min impregnation process	beef	[19]
red pepper, garlic, onion, red pepper, tomato, pepper, garlic, or pepper, red pepper, and garlic	24 h at 4 °C	beef	[26]
olive oil and rosemary, pumpkin oil and fresh oregano, sunflower oil and thyme, walnut oil and fresh basil, sesame oil and ginger plant	120 h at 4 °C	beef	[27]
water, 2% salt, 0.5% sugar, 0.5% soy sauce, and spices: paprika, clove root, ladybug anise, tangerine, long pepper Chinese cinnamon, muscatel spice, trifoliate orange, fennel, Dahurian angelica, Cinnamomum cassia, liquorice, green cardamom, hawthorn	2 h tumbling at 4 °C; vacuum.	beef	[40]
sugar, onion, water, lemongrass, salt, turmeric, cinnamon, coriander, and oil	0, 4, 8, and 12 h at 4 °C (immersion)	beef	[58]
(I) seasoning (marjoram, thyme, lemon pepper, oregano, basil, and garlic powder); (II) lemon and honey marinade (pH 4.8) (honey, fresh lemon juice, soy sauce, black pepper, and water); (III) balsamic vinegar marinade (pH 4.4) (light soy sauce, Dijon mustard, balsamic vinegar, water, brown sugar, and black pepper); (IV) pineapple marinade (pH 4.1) (fresh pineapple juice, soy sauce, balsamic vinegar, water black pepper, red pepper flakes, and garlic powder); (V) ginger marinade (pH 4.7) (soy sauce, lime juice, fresh grated, ginger, water, and crushed red pepper).	24 h at 4 °C	deer, wild boar	[71]
ginger	1 h tumbling at 4 °C	chicken	[164]
garlic	24 to 48 h at 4 to 7 °C	pork	[165]
soy sauce and hot pepper paste	overnight at 4 °C	pork	[166]
oregano, liquid smoke (as base and salt, phosphate, nitrate, soy sauce, meat broth, black pepper, lemon pepper, cayenne pepper, red curry paste, mild Tabasco, mustard, fructose, xylose, honey, garlic powder, extract from onion, oregano and paprika, tomato purée, lemon juice, lime juice, cognac aroma, fermented milk, beer, and bacon aroma)	not stated	pork	[167]
spices and flavourings, salt, and oil	24 h at 4 °C	pork	[168]
nanoparticle paprika oleoresin (1 and 3 g/100 mL) and water/milk	tumbling for 20 min at 4 °C	poultry	[169]
tomato paste, red pepper paste, sunflower oil, red pepper, black pepper, cumin, salt, fresh lemon juice, and garlic rind	24 h at 4 °C; stored 1–10 days	chicken	[170]
lacto-fermented garlic	3 days at 4 °C	lamb	[171]
thyme, rosemary, basil, marjoram, cinnamaldehyde, linalool, and lactic acid	7 days at 4 °C	chicken	[172]
turmeric, curry leaf, torch ginger, and lemongrass	8 h at 4 °C; PE bags	beef	[173]
coriander leaf extract and coriander root extract	4 h at 4 °C	duck	[174]
Marinade 1 (45.5% pomegranate syrup, 23% water, 17% honey, 11.5% mustard powder, 2% NaCl, and 1% pepper; pH 2.45); Marinade 2 (73% lemon juice, 18% honey, 2% garlic, 2% NaCl, and 1% pepper; pH 2.10); Marinade 3 (52% white wine vinegar, 24% sugar, 2.5% estragon, 18.5% onion, 2% NaCl, and 1% pepper; pH 2.87).	not stated	chicken	[175]
341 mL beer, 1 g oregano, 1 g parsley, 4 g mustard, 2 g salt, 8 g pepper, 1 g garlic, 25 mL olive oil, 15 mL vinegar, and 25 g fresh onions	12 h at 4 °C; Ziploc closed plastic bags.	beef, moose	[176] [177]
sodium chloride, 3% of a commercial blend of polyphosphates; EO mixture (1:1) of thyme and orange	20 min tumbling (20 rpm) at room temperature; vacuum (78 kPa)	chicken	[178]
hibiscus extract	not stated	beef	[179]
dry ground marjoram and thyme, garlic powder, fresh horseradish, lime tree honey, and red wine	12 days at 4 °C	beef	[180]
fresh turmeric, torch ginger flower, curry leaves, and lemongrass	8 h at 4 °C 24 h at 4 °C 24 h at 4 °C	beef lamb beef	[6] [181] [182]
cinnamon powder, and green tea powder	not stated	pork	[183]
salt, white pepper, and garlic powder	not stated	pork	[184]
sage leaf (*Salvia fruticosa*), hops (*Humulus lupulus*), licorice root (*Glycyrrhiza uralensis*), curcuma (*Curcuma xanthorrhiza*), clove bud (*Syzygium aromaticum*), oregano leaf, and ajowain seed (*Trachyspermum ammi*)	not stated	chicken	[185]
red pepper powder, red pepper seasoning (red pepper powder, sea salt, garlic, and onion)	24 h at 5 °C	pork	[186]
rosemary, sage, and thyme	7 days at 4 °C	turkey	[187]
soy sauce, pepper, garlic, oregano, rosemary, and chili	5 h at 4 °C	beef, pork	[188]
black pepper, garlic, salt, canola oil, and aromas	24 h at 4 °C	beef	[189]

Spices and herbs in marinades can have a significant impact on the quality of meat in terms of its flavour, tenderness, and overall quality [190,191,192] and can help balance the flavours of marinade ingredients (e.g., chili powder or black pepper balanced with sweet ingredients such as honey) and harmonize the taste profile. They are responsible for enhancing the meat’s flavour and creating the unique sensory profile of the meat [67]. Marination with spices and herbs has been used to improve the functionality and safety of meat [40]. Nassu et al. [193] reported that an antioxidant, such as rosemary, retarded the development of oxidized aromas and flavours. Spices are central to marinades as they provide a wide range of flavours and aromas [40,127,191,194,195,196,197].

Common spices such as paprika, pepper, chilli powder, garlic, ginger, cumin, and turmeric can add depth and complexity to the meat’s taste [40,127,184,185,191,192,195,197,198]. In meat marinades, pepper (black, white, and green) is a popular spice often used to add flavours, aromas, and a hint of spiciness [184,196]. The selection between types of pepper depends on one’s culinary preferences and the specific flavour to be obtained. Each type has its unique characteristics that can enhance different types of meat and other products. In addition, black pepper (*Piper nigrum*) is enriched with phenolic compounds [198], mainly piperine, an alkaloid which is responsible for the pungency of black pepper. The antioxidant properties of the polyphenolic compounds contained in black pepper have been confirmed in studies conducted on beef hamburgers. The addition of black pepper had a significant effect, lowering the malondialdehyde (MDA) content of the hamburgers studied compared to that of the control group. In addition, this study demonstrated the synergistic effect of black pepper on turmeric, as their combination significantly reduced the MDA values in the samples, compared to those of the spices used alone [198].

The use of paprika in the form of nanoparticle paprika oleoresin in a marinade carrier system, as a function of the water-to-milk ratio (water, milk, and NaCl), had a significant impact on the sensory quality of chicken [169]. It was observed that the amount of paprika had a significant effect on all sensory colour attributes (surface orange and red intensity and colour penetration). The most beneficial effect was shown with 3 g/100 mL of paprika in the marinade. This resulted in a cooked meat product with a higher red and orange intensity as well as deeper colour penetration. A similar relationship was shown for paprika’s flavour and overall acceptability. However, considering the paprika carrier in the marinade, the highest colour penetration was observed for the water-based carrier (100:0 ration of water: milk). In contrast, the highest absorption of the marinade was shown for the milk-only carrier system (0:100 ratio of water to milk) [169]. The use of a marinade of onion juice or garlic also had a positive effect on the juiciness and tenderness of pork meat [82].

Studies have shown a significant effect of herb- and spice-based marinades on the sensory quality of meat, as well as its flavour, aroma, and freshness [27,67,176,177,199]. Marinades based on turmeric and lemongrass (52.42%:47.57%) significantly improved the colour and flavour as well as the overall quality of grilled beef samples [173]. In addition, marinades with both aromatic herbs and cold-pressed oils have a positive effect on beef tenderness and juiciness after longer marination times. The use of herbal and spice extracts also causes changes in the pH of the meat, limiting its rise. This is reflected in the slowing down of the process of protein degradation, thus extending the meat’s shelf life [200].

The addition of spice extracts, such as bay leaf, black pepper, turmeric, jalapeno pepper, and tamarind paste, to marinades increased the proportion of the colour components L* and b* and lowered the hardness and pH of the meat [201]. However, herbs are a common ingredient in meat marinades not only for their flavourful qualities. They are also used because of their antioxidant [143,199,202,203,204] and antimicrobial properties [66,143,178,205]. Rohod et al. [204] demonstrated that rosemary and oregano have similar efficiencies to synthetic antioxidants such as BHT in inhibiting the development of *Staphylococcus aureus* in marinated chicken breast. The use of marinades with thyme and orange blends effectively reduced lipid oxidation in chicken meat, without a negative effect on the meat’s sensory quality or colour parameters [178]. Studies conducted by Pathania et al. [35] also showed that teriyaki and lemon pepper marinades could both reduce the *Salmonella* load on chicken skin and red meat during aerobic storage.

Natural antioxidants, such as the phenolic compounds in extracts, can contribute to the inhibition of cyclisation and oxidation reactions by quenching or scavenging free radicals, thereby increasing the safety and shelf life of grilled meat products [201]. The herbs and spices used in marinades improved the meat quality by affecting the pH, textural properties, colour, PAH profile (reducing the formation of PAHs), and volatile compounds in grilled meat [174,201,206]. Using an oolong tea infusion (1%) for a marinade, Caliskan et al. [81] achieved a 94.4% reduction in the total heterocyclic aromatic amine (HAA) content in chicken meat, and similar findings were recorded by Gibis and Weiss [179] regarding the use of hibiscus extract. Combinations of spices, such as turmeric, lemongrass, torch ginger, and curry leaves, also effectively inhibit the formation of HCAs in grilled beef [6,181] and also in lamb meat [182]. Lai et al. [183] showed that the addition of cinnamon powder (0.5%) or green tea powder (0.5%) reduced HAA and PAH formation in marinated pork.

A study by Pathania et al. [35] showed the positive effect of using a lemon pepper marinade (ground black pepper, lemon peel granules, lemon oils, and some extracts of spices) in reducing the growth of *Salmonella* bacteria in chicken meat. This has to do with the lower pH of the lemon pepper marinade due to the presence of lemon oils, which resulted in more effective antimicrobial activity [35]. Furthermore, the use of a marinade with a suitable composition of spices can effectively exhibit antimicrobial activity against *Campylobacter jejuni and Enterobacteriacea*, as well as provide desirable organoleptic characteristics in chicken meat [175]. 

Oregano essential oil has been found to display antimicrobial activity against pathogenic microorganisms, such as *Escherichia coli, Listeria monocytogenes,* and *Salmonella enteritidis,* in beef and pork meat [163]; rosemary essential oil (0.05%) was found to be able to inhibit the growth of *Listeria monocytogenes, Escherichia coli,* and *Staphylococcus aureus* in beef and chicken meat [204,207,208]; and thyme essential oil (0.08%) added to meat inhibits the growth of *Pseudomonas* spp. and *Staphylococcus aureus* [163].

The addition of essential oils improved the sensory quality and increased the overall acceptability of samples, especially at the end of storage, and was preferred among consumers [209,210,211,212,213]. Essential oils, including rosemary, thyme, oregano, basil, coriander, ginger, garlic, clove, juniper, and fennel, are characterised by their strong antioxidant activity [214,215]. Essential oils used alone or in combination have been shown to extend the shelf life of meat and meat products, and this has been associated with a reduction in lipid oxidation [209,211,212,216]. It is worth mentioning that due to the low sensory thresholds characteristic of essential oils [161], their sensory compatibility and impact on other ingredients on the sensory profile of the final product should be carefully considered [217,218].

### 3.4. Effect of Marinades Based on Fermented Dairy Products on the Quality of Meat

Natural fermented dairy products (FDPs) such as kefir, yoghurt, buttermilk, acid whey, and acid milk are also used as marinades (Table 4).

FDPs have a low pH (4.6) and contain live cultures of microorganisms, mainly lactic acid bacteria (LAB). Fermentation processes of dairy products generally increase the nutritional value and bioavailability of nutrients [227], and the action of certain strains of LAB can lead to the removal of milk components such as lactose and galactose [228]. The conversion of lactose leads to the production of lactic acid in the fermented product. LABs also exhibit casein proteolytic activity and cause the release of amino acids and peptides. In addition, bacterial enzymes convert milk carbohydrates into oligosaccharides, some of which have prebiotic properties [229].

Buttermilk (BM) is rich in polar lipids, including phospholipids and sphingolipids. It has lower concentrations of neutral lipids, such as mono-, di-, and triglycerides, as well as cholesterol and its esters [230]. Additionally, BM contains lecithin and a small amount of protein with high biological value [69]. Acid whey (AW), a by-product of cottage cheese and yogurt (YO) [14,224] production, is a source of whey proteins (α-lactalbumins, β-lactoglobulins, lactoperoxidase, serum albumins, lactoferrin, and immunoglobulins), vitamins B and A, significant amounts of tryptophan (a serotonin precursor) and cysteine (a glutathione precursor), and minerals. These products are a valuable source of lactic acid bacteria (LAB) and have antimicrobial and antioxidant activities. They also help to limit the addition of chemical preservatives to food [175,231]. In the process of the fermentation of kefir (KE), in addition to LAB, yeast is also produced. KE contains fermentation products and metabolites such as kefiran and exopolysaccharides, which have additional health benefits [227,232]. 

The results of studies [3,25,220,225,226,231,233] show that the presence of live LAB cultures in FDP can extend the shelf life of different types of marinated meat, inhibit oxidative processes, improve colour stability after heat treatment, and also improve the physicochemical properties of meat products. 

Many authors report the beneficial effect of acid marinades on the sensory quality of meat products [3,30,191,219,222]. For example, marinating pheasant breasts in AW and BM significantly increased their juiciness and tenderness; additionally, BM improved the taste, and AW significantly reduced the intensity of the specific meat aroma [69]. Wild boar meat marinated for 7 days in KE was characterised as having high levels of tenderness, juiciness, and overall attractiveness compared to those of meat marinated in calcium chloride, wine, or a pineapple marinade [219]. Extending the marination time of meat in FDP had a negative impact on its sensory characteristics. In the study by Latoch et al. [220], pork loin marinated for 3 days in FDP had a higher sensory rating than when it was marinated for 6 days. The best ratings were given to meat marinated in BM. Similarly, extending the marinating time of chicken breasts in BM or AW from 24 to 48 h had a negative impact on the intensity of the flavour of heat-treated products [222]. 

Instrumental measurements of texture parameters are one of the most frequently performed analyses of meat products marinated in FDP, and many authors confirm the beneficial effects of this process [29,30,224]. Marinating meat at low temperatures (0–4 °C) causes its further, although limited, maturation. Proteolysis leads to the structural weakening of myofibrils as well as the endomysium and perimysium of connective tissue, resulting in increased meat tenderness [234]. 

The addition of calcium-containing ingredients, such as kefir, has been shown to increase proteolysis caused by calpains [78,219]. Żochowska-Kujawska et al. [78,219] showed that marinating wild boar and deer meat in KE for several days (4 or 7 days) causes structural changes, such as increasing the cross-sectional area of the muscle fibre, improving its shape, and reducing the thickness of connective tissue. This reduces the meat’s hardness and springiness and increases its tenderness and juiciness.

Marinating pork for 6 or 9 days in BM or YO reduced its hardness and chewiness and, regardless of the type of marinade, reduced its cohesiveness. There was no effect of the type of marinade or the marinating time on the springiness of the meat [7,25,220,221]. However, other studies [14,224,235] prove that marinating pork, lamb, and rabbit meat in AW for too short a time, i.e., for 5 h, does not affect its tenderness. However, marinating the meat for 10–24 h improves its tenderness (reduces the cutting force), without negatively affecting its other quality features. At the same time, these studies did not demonstrate the effect of a longer marinating time on the tenderness of chicken meat. The authors attribute the different levels of effectiveness of AW as a tenderizer in pork and poultry to the fact that acidic marination seems to be more effective in tenderizing muscles with a high content of connective tissue. This was not confirmed by other authors [3,69,70,175,191,222], who found that marinating the breast muscles of chickens, pheasants, and turkeys in AW or BM for 12–48 h reduced the cutting force, springiness, and chewiness.

Many authors [7,25,69,220,221] have found no influence of the type of FDP marinade on the colour lightness (L* parameter). Other authors [14,175,191,222,224] have noticed a lightening of the colour of marinated meat, both raw and heat-treated. 

Proteolysis by meat proteases (calpains and cathepsins) and AW-derived endopeptidases may affect the meat’s water-binding capacity. Water introduced into the meat during marination may affect its ability to reflect light from its surface [29]. In the case of cooked pork loin marinated in FDP, other authors [7,25,220,221] found a darkening of colour, which may be the result of water loss during cooking [236] or may result from the presence of minerals and sugars in FDP or those added during marination [69]. Wójciak et al. [226] hypothesized that β-lactoglobulin, which is the main AW protein, may serve as a source of glutathione and thiol amino acids, which protect meat products against discoloration.

In the case of heat-treated meat products, the intensity of redness is inversely proportional to the degree of myoglobin denaturation during marination and processing. Karageorgou et al. [224] found that the redness (a*) was reduced by marinating pork and chicken meat in yogurt whey in both raw and cooked meat. This was confirmed by other studies with FDP marinades, but this effect depended on the marination time and cooking temperature [3,7,25,191,220,221]. Wójciak et al. [226] demonstrated the influence of AW on the formation and stabilization of the pink/red colour of uncured products. Similarly, Augustyńska-Prejsnar et al. [69,175,222] showed the effect of marination with AW on the redness of heat-treated chicken and pheasant breasts.

The beneficial effect of marination on the redness (a*) of meat can be explained by the fact that the hydrolysis of milk proteins can generate bioactive peptides with antioxidant properties [237]. An important factor shaping the colour of meat is the redox potential, which determines the redox status of the iron located centrally in the porphyrin ring of the myoglobin molecule [238]. A low redox potential also helps keep heme pigments in a reduced form. The use of FDP can limit the oxidation of myoglobin and increase its thermal stability, thus minimising changes in the a* parameter and increasing redness [7,25,220,221]. The presence of microflora in AW leads to higher redox potential values [226].

The influence of marinating meat in FDP on the b* parameter is not clear. Some authors [7,25,220,221,225] have found no effect of FDP on the value of the b* parameter in raw and heat-treated meat. Other studies [69,175,191,222] indicate a reduction in the level of yellowness after marinating meat in BM and AW. In turn, other authors [14,224] have found an increase in yellowness in raw chicken meat marinated in yogurt whey, but they did not observe significant differences in samples after heat treatment. The same authors noted a decrease in the value of the b* parameter in raw pork marinated in yogurt whey. 

Marinating meat in FDP does not affect the oxidation of fats in raw material, but it prevents [14,25,221,224] or has little effect [225] on fat oxidation in finished meat products. This is due to the fact that LAB do not have lipase activity [232], and some LAB have antioxidant effects and are able to reduce the risk of the accumulation of reactive oxygen species [239]. Additionally, LAB can decompose superoxide anions and hydrogen peroxide. Peptides with antioxidant activity have been identified in FDP [240], including casein calcium peptides [241]. Diaz and Decker [242] report that cooking increases the catalytic activity of iron in meat, while milk proteins have a strong chelating effect, which can also effectively limit oxidation.

Kęska et al. [243] showed that AW contributes to the production of peptide fractions in meat products, which may have a beneficial physiological effect on the human body. AW, which is rich in LAB, is an additional source of enzymes, facilitating the extraction of more peptides and free amino acids in marinated meat. The action of LAB in meat results in the better colonization and better activation of LAB enzymatic mechanisms, including proteases [243]. Research by Wójciak et al. [225] showed that beef marinated in AW for 48 h contained more peptides than beef marinated for 24 h. The addition of AW, probably due to the presence of *Lactobacillus plantarum* in the whey, which has a proven ability to produce amine oxidase enzymes and degrade biogenic amines, reduces the amount of biogenic amines [244].

Acidic marinades effectively inhibit the growth of microorganisms [245] due to their low pH and the substances contained in FDP, such as organic acids, LAB, and other metabolites produced by LAB (including whey proteins, bacteriocins, and polyphenols) [1,3,69,70,175,226]. A large amount of LAB in raw meat guarantee its microbiological stability and product safety.

Factors affecting the number of microorganisms in meat products, in addition to the concentration of hydrogen ions, may be the water activity, the presence of oxygen, the redox potential of the environment, the activity of enzymes of microbial origin, and the presence of compounds and microflora that inhibit the development of specific groups of microorganisms [246]. Brik and Knøchel [175] found that yoghurt used for marinating pork was not as effective as red wine in reducing bacterial viability, but *C. jejuni* is extremely sensitive to YO compared to other bacteria. The use of buttermilk and acid whey as a marinade for meat increases the microbiological safety of the product compared to that of a product marinated in lemon juice, while maintaining its good technological features [70]. It significantly reduced the number of mesophilic aerobic bacteria, *Pseudomonas* spp., and bacteria from the *Enterobacteriaceae* family in raw chicken, pheasant, and turkey meat. According to the authors, the main factor influencing the reduction in microorganisms in this case was the lowering of pH, caused mainly by the presence of lactic acid, but the type of marinade does not affect the pH of the meat [14,29,30,69,70,191,221,222,224].

Eldaly et al. [247] investigated the effect of a yoghurt-based marinade on the levels of five PAHs in grilled beef (kebab and kofta). In addition to yoghurt, individual ingredients, such as salt, turmeric, curry, cardamom, vinegar, mustard, and onion, were examined. Marinating the meat before grilling it reduced the PAH levels to 57.93 µg/kg in grilled kebab (about 50.6%) and 30.2 µg/kg in grilled kofta (about 49%).

### 3.5. Effect of Marinades Based on Other Fermented Drinks on the Quality of Meat

Nowadays, alcoholic beverages, such as wine and beer, can also be used for the marination of meat [67]. Examples of the use of wine and beer for marinating meat are presented in Table 5.

Beer is used to season meat and used as a marinade, alone or in combination with other spices, to counteract oxidative processes and limit the formation of harmful compounds, such as HAAs and PAHs during high-temperature thermal treatment. Beer is a natural food ingredient rich in antioxidants that can protect meat lipids from oxidation [259]. Antioxidant activity results mainly from the presence of phenolic compounds that are capable of donating hydrogen radicals and pairing with lipid radicals formed at oxidation [103].

Unfiltered beers, compared to filtered beers, are generally richer in antioxidants and polyphenols, such as hydroxybenzoic acid and hydroxycinnamic acid. Research by Manful et al. [248] showed that marinating meat in these beers significantly inhibited the formation of HAAs during grilling. India dark beer contains antioxidant melanoidins [260], and wheat beers contain lemon and lime, which are a source of antioxidant limonoids and vitamin C [248]. In many studies [248,249,259,261], it has been shown that beer marinades in combination with herbs have an inhibitory effect on the oxidation of functional lipid components in grilled meats, such as the following: phosphatidylcholines and phosphatidylethanolamines, plasmalogen, fatty acids, diglycerides and monoacetyl diacylglycerides, conjugated fatty acids, and the formation of HAAs.

Beer-based marinades increase the total antioxidant activity, total phenolic content, and total oxidized terpene content, and they reduce the total oxidant content in marinated grilled elk meat and beef. It is also an effective technique for the protection of MUFA- and PUFA-enriched plasmalogens. However, India-ale-based marinated meats were found to be more effective compared to wheat-ale-based marinated meats [176,177,248,249,250]. Herbal and spice marinades based on unfiltered beer improve the quality, safety, and sensory attributes of grilled elk and beef meats, in line with the preferences of consumers [249]. Marinating meat reduced the number and content of volatile compounds originating from lipid oxidation processes and reduced the content of Maillard reaction products, especially pyrazine, which is a precursor of some heterocyclic amines. Consumers preferred meats marinated with beer rather than unmarinated ones, and sulphur derivatives and volatile terpenes resulted in better evaluations of aroma and taste.

Żochowska-Kujawska et al. [78,219] showed that marinating wild boar and deer meat for 4 or 7 days in dry red wine and then cooking them reduces their hardness and springiness and increases their tenderness and juiciness. Other authors [103,252] have observed that the use of beer- and wine-based marinades for marinating grilled pork reduces the amount of thermal loss, depending on the heat treatment time.

Studies conducted using red wine marinades [103], white wine [261], and Pilsner beer [103,261] have indicated that the strongest inhibitor of HAA formation in fried beef was beer marinade (Pilsner beer), reducing their level by over 80% compared to those of unmarinated meat.

Although beer marinades have lower antioxidant activity compared to wine marinades, their inhibition of HAA formation is more effective. Sugar and dextrins present in beer may also have a stronger inhibitory effect [103]. Fried, beer-marinated beef also received high notes in the sensory evaluation. Similar results were achieved by Viegas et al. [260] who investigated the effect of marinating pork in different types of beer on the formation of PAHs in charcoal-grilled pork. Black beer showed the greatest effect, reducing the PAHs in meat by approximately 90%. A non-alcoholic pilsner was less effective than one containing alcohol, which reduced PAHs by about 58% and 70%, respectively. Important components of marinades are additives, which influence and enhance the taste of meat, especially spices. A strong positive correlation was observed between the inhibitory effect of beer on the formation of total HAAs and its antioxidant activity [251].

Vidal et al. [249] found that the use of beer and wine in marinades increased the PAH content in grilled meat products. The authors of [252] suggest that this may be due to the high content of polyphenolic compounds contained in wine, such as anthocyanins and tannins, which play an important role in the formation of Maillard compounds and constitute an additional organic substrate subject to pyrolysis. The use of aluminium trays in the grilling process, constituting a barrier between the fire and the raw material, reduced the content of benzo[a]pyrene to a level below the detection limit of the method, and the PAHs were reduced by 86.3% and 70% in grilled meat marinated in wine and beer, respectively [252].

Wine, mainly red, is commonly used as an ingredient in marinades. Wine consists mainly of water, ethanol, organic acids, sugars, pigments, and various aromatic components, such as polyphenols, but it is a complex product containing over 600 substances [175]. There are several studies on which wine fraction (organic acids, ethanol, or polyphenols) is responsible for specific bioactive activities. Wine has a low pH (3.0–3.6) due to the presence of organic acids such as tartaric, malic, succinic, lactic, and acetic acids [262]. The high concentration of wine may be responsible for a significant reduction in pH, but at the same time, it provides meat products with antioxidants (phenolic compounds). Research by Brik and Knøchel [175] indicates that marinating meat in wine can improve its taste and texture as well as ensure safer food. The addition of wine can inhibit microbial growth and delay the oxidation of lipids and proteins [5,263]. 

The varietal diversity of wines, and therefore their composition, may influence the quality parameters of marinated meats in various ways. For example, Arcanjo et al. [84] found that Carbernet and Tempranillo wines, rich in procyanidins, are more effective against lipid oxidation than Isabel wine, rich in hydroxycinnamic acids (mainly caftaric acid), which better protect proteins against oxidation.

Compared to white wines, red wines have similar or slightly stronger antimicrobial effects [264]. In several studies [36,257,265], it has been shown that red wine has a pronounced antibacterial effect against *Salmonella enteritidis*, *Escherichia coli*, *Listeria monocytogenes Vibrio parahaemolyticus*, *Shigella sonnei,* and *Helicobacter pylori*. Individual organic acids typical of wine, such as malic acid and tartaric acid, showed inhibitory effects on *Escherichia coli*, *Listeria monocytogenes*, *Staphylococcus aureus* and *Salmonella typhimurium*, and a synergistic effect of the combination of ethanol, organic acids, and a low pH was observed [264]. 

Just and Daeschel [266] showed that pathogenic bacteria survive much longer in grape juice than in red wine despite having the same pH, which indicates the influence of the type of acid and other metabolites formed during fermentation. Arcanjo et al. [84] showed that the specific phenolic composition and high organic acid content of Isabel wine may explain its effect on *Enterobacteriaceae*, while the presence of sugars in Carbernet and Tempranillo wines may promote the growth of LAB in wine-marinated beef. McKee et al. [267] reported that using red wine as a rinsing agent for chicken breasts significantly reduced the total counts of aerobic bacteria and coliforms.

Research by Brik and Knøchel [175] showed that marinating meat in red wine has an antibacterial effect on *B. thermosphacta* and *C. jejuni* and, to a lesser extent, on *C. maltaromaticum* and *L. monocytogenes*. The microbial procedure was more effective when the meat was immersed in red wine at 42 °C for 15 min before storage at 4 °C than when the meat was immersed in red wine at 4 °C throughout the experiment. 

Marinating beef in red wine can increase the shelf life by reducing the total live count (TVC) and the *Pseudomonas* spp. that cause spoilage in raw meat [5].

Mantzourani et al. [254] showed that wine-based marinades containing the ethanol extract of pomegranate (*Punica imprezaum* L.), alone or in combination with thyme and oregano essential oils, increase the resistance of pork to spoilage. Even though the antibacterial or bactericidal activity of wine has been reported by scientists, its exact mechanism has not been explained [254].

A positive correlation was found [253] between the antioxidant properties of wine and the content of some HAAs. Red wine marinades reduce the formation of certain HAAs. However, the marinating time had a variable effect on the formation or inhibition of individual HAAs in wine-marinated fried chicken. Melo et al. [103] found that although wine marinades have higher antioxidant activity compared to beer marinades, the inhibition of HAA formation is less effective.

Antioxidant phenolic compounds in red wine can prevent the oxidation of heme pigments, but increasing the addition of wine to 10% did not enhance this effect. Istrati et al. [180,256] found marination had no effect on colour parameters, but their treatment had a significant impact on the percentage of Mb (myoglobin), MMb (metmyoglobin), and MbO2 (oxymyoglobin). The oxymyoglobin content increased after 14 days of marination, while the metmyoglobin level decreased.

Marinating beef in wine-based marinades with the addition of spice plants significantly reduced thermal losses and increased tenderness [180,256], which is related to changes in myofibrillar proteins. The electrophoretic pattern of muscle proteins showed a decrease in the molecular weight and relative density (%) of some protein fractions for marinated meat compared to those of a control sample. The mechanism of the tenderizing effect of acidic marinades, such as dry wine, involves several factors, including the weakening of structures due to the swelling of meat, increased proteolysis by cathepsins, and increased conversion of collagen to gelatin at low pH levels during thermal treatment [87]. The marinades affected connective tissue and myofibril proteins, causing an increase in the content of protein nitrogen, free amino acids, and hydroxyproline in cooked cuts of beef. Żochowska-Kujawska et al. [78,219] found that 4 or 7 days of marinating wild boar and deer meat in dry red wine slightly reduced the cross-sectional area of the fibres and the thickness of the endomysium and perimysium of the connective tissue, causing a significant reduction in hardness and an increase in tenderness and juiciness.

### 3.6. Effect of Marinades Based on Natural Ingredients on Meat Quality

In order to summarize the impact of marination with marinades based on natural ingredients, Figure 2 was prepared.

## 4. Conclusions

Marinating meat is a traditional process to improve the quality of meat (Figure 1). However, recently, due to the “clean label” trend, natural ingredients are being used instead of synthetic ingredients. In this review, they have been divided into four groups according to their origin: plant-based marinades; marinades based on seasonings; fermented dairy products; and other fermented drinks. The published research results show that the following are important: the proper selection of marinade ingredients for a specific type of meat, the marination method and conditions (time, temperature, and ratio of meat to marinade), and knowledge of the interactions between the ingredients in marinades. Marinades based on natural ingredients improve meat’s sensory quality, culinary properties, shelf life, and safety. This is possible due to the chemical compounds naturally present in marinade ingredients, such as the following: organic acids, phenolic compounds, and plant exogenous proteolytic enzymes, which are acceptable to consumers.

The addition of acidic, natural marinades increases the tenderness of the meat and thus makes it easier to consume, especially by the elderly. The utilization of natural ingredients may also enhance consumers’ willingness to buy, in light of the recent increasingly popular attitude towards the consumption of ‘clean-label’ products.

The gaps and future trends of this topic were identified. The literature lacks research results on the impact of marinades with the addition of natural ingredients on the nutritional value and biological potential of marinated meat products and dishes, including their health-promoting (functional) aspects. Limited data indicate a negative impact on sensory quality. Future research will probably include studies on the impact of marination with the use of other, so far unused natural ingredients, and the use of post-production waste for the composition of marinades, e.g., fruit pomace, which will reduce food waste and are in line to the “no waste” trend, while also potentially reducing marinade production costs.

## Figures and Tables

**Figure 1 foods-12-03638-f001:**
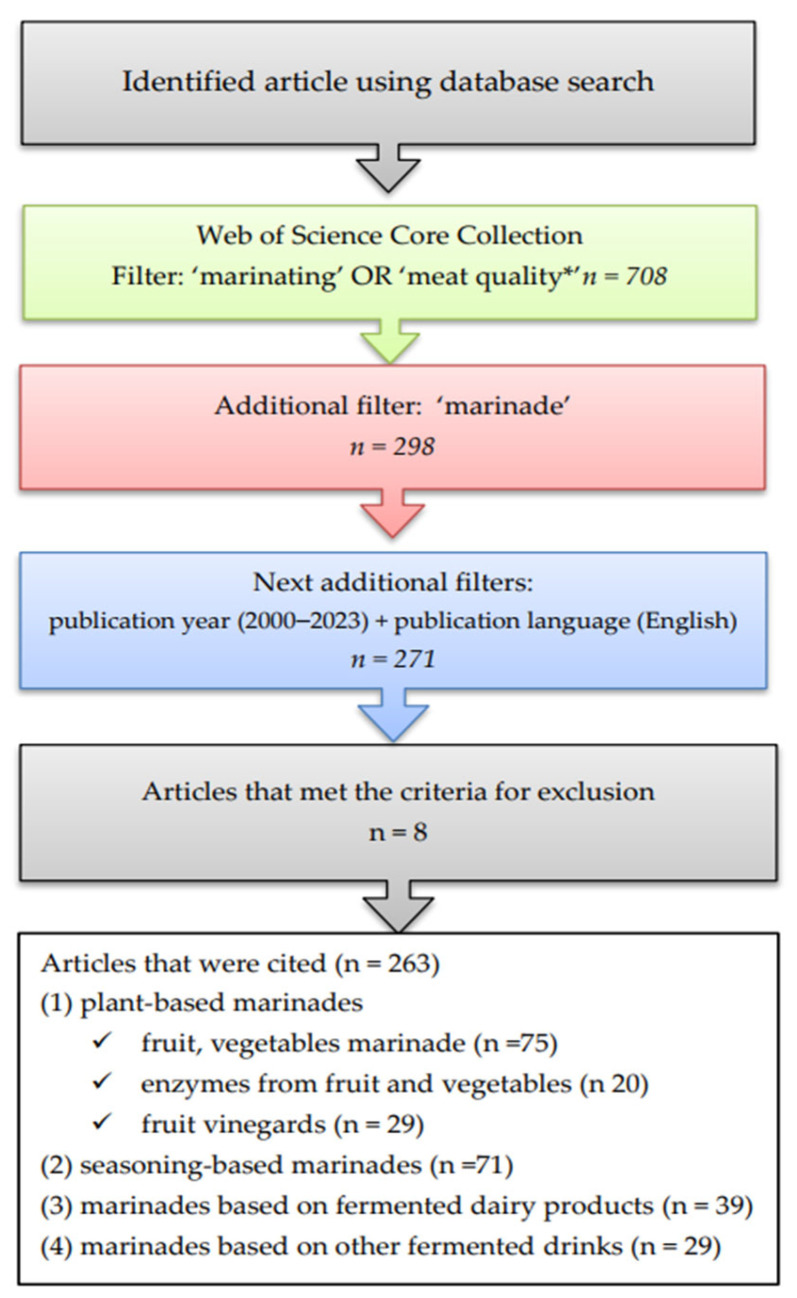
Procedure for searching and selecting manuscripts for review.

**Figure 2 foods-12-03638-f002:**
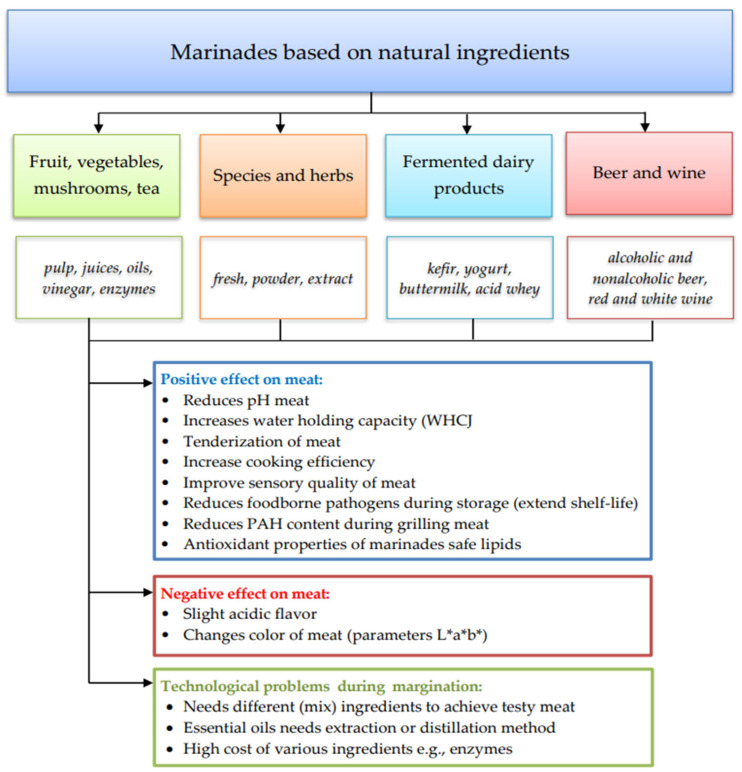
Natural marinades and their effect on meat quality.

**Table 1 foods-12-03638-t001:** Selective examples of plant-based marinades.

Marinade Based on	Marinating Condition	Meat	Source
beetroot	2 h at 4 °C	chicken	[16]
koruk juice	1, 2, and 18 h at 4 °C	poultry	[21]
mango or pineapple juice June plum juice (tumbled)	12 and 24 h at 4 °C	chicken	[62]
sour cherry and plum juice	24 h at 4 °C, 1:1 (meat: marinade)	pork	[63]
yellow mombin (50, 75 or 100% pulp with water)	7, 14, and 21 days at 4 °C	pork	[64]
lemon juice	0, 4, 8, 12, 24, 36 and 48 h at 4 °C various ratio meat: marinade	beef	[65]
(I) lemon juice (100%, water, lemon juice and oil, sodium bisulfate, sodium benzoate, pH = 2.58) (II) tomato juice (water, tomato concentrate, salt, ascorbic acid, pH 4.03), soy sauce (water, wheat, soybean, salt, sodium benzoate, pH 4.59)	48 h at 4 °C 1:1 (meat: marinade)	chicken	[66]
lemon juice (70%), pomegranate juice (70%) or mix in ratio (35% + 35%) +olive oil, honey, thyme	1, 3, 6, and 9 h at 4, 10, and 20 °C	chicken	[67]
lemon juice (5%, 10%), water/olive oil	3, 9 and 15 days at 4 °C, 1:1 (meat: marinade)	pork	[68]
lemon juice (pH 4.52)	24 h at 4 °C, 1:1 (meat: marinade)	pheasants	[69]
lemon juice	12 h at 4 °C 1:2 (meat: marinade)	turkey	[70]
lemon and honey marinade (pH 4.8) pineapple (with seasonings, pH 4.1)	24 h at 4 °C	jerky roe deer, wild boar	[71]
lemon juice, olive oil, dried thyme, salt	1 h 4 ℃ or 20 ℃, 98.81% vacuum or atmospheric pressure	pork	[72]
citrus juice (31% orange juice, 31% lemon juice, and 38% water)	20 h at 4 °C marinade (10 ml/g meat)	beef	[73]
black currant juice	24 h at 5 °C	pork	[74]
black mullbery juice (pH 3.96) grape juice (pH 3.86), pomegranate juice (pH 3.19)	72 h at 4 °C	chicken and turkey	[75]
koruk juice and dried koruk pomace	2, 24, 48 h at 4 °C	beef	[76]
blueberry, raspberry, and strawberry	1, 6, 12, 24 h at 4 °C	camel, beef, chicken	[77]
pineapple juice	4 days at 4 °C	wild boar, deer	[78]
fruit juices (chaenomeles and cranberries)	not stated	pork, beef	[79]
asparagus juice	4, 24, 48 h at 4 °C	beefsteak	[80]
white tea and oolong tea leaves	16 h at 4 °C (immersing)	chicken	[81]
onion juice, garlic juice (3%, 6%)	24 h at 4 °C	pork	[82]

**Table 4 foods-12-03638-t004:** Selective examples of marinades based on fermented dairy products (FDPs).

FDP Used to Marinade	Marinating Condition	Meat	Source
kefir	4 days at 4 °C; vacuum	wild boars and deer	[78]
kefir	7 days at 4 °C; vacuum	wild boars	[219]
kefir, yoghurt, or buttermilk	3, 6, 9, or 12 days at 4 °C; vacuum	pork	[7] [25] [220]
kefir, yoghurt, or buttermilk	48 h at 4 °C, vacuum	pork	[221]
buttermilk or acid whey (salt, cane sugar)	24 h at 4 °C 24 and 48 h at 4 °C 12 h at 4 °C	pheasant chicken turkey	[69] [70] [222]
buttermilk or sour milk	12 h at 4 °C	chicken	[191]
yoghurt	3 days at 4 °C	pork	[175]
yoghurt acid whey	20 h at 4 °C	pork, lamb, rabbit, chicken	[14]
acid whey (salt, cane sugar)	24 h at 4 °C	chicken	[223]
yoghurt acid whey (with or without hesperidin)	5 h at 4 and 20 °C	pork	[224]
	10 or 15 h at 4 °C	pork and chicken	[225]
acid whey (salt, glucose)	24 h at 4 °C	beef	
acid whey (salt or salt, mustard)	immediately before heat treatment	pork (sausage)	[226]
acid whey (salt)	12 and 24 h at 4 °C	poultry	[3]

**Table 5 foods-12-03638-t005:** Selective examples of marinades based on beer and wine.

Beer and Wine Used to Marinade	Marinating Condition	Meat	Source
dry red wine (with or without oregano essential oil and garlic powder, dried onion, and freshly crushed black pepper)	12 h at 5 °C and 10 days at 5 °C or 5 days at 15 °C	beef	[5]
red or white wine (10%)/olive oil beer (10%)/olive oil beer/lemon juice/olive oil	1:2/1:3 (meat: marinade) 1:2/1:3 1:2:1/1:1:1	pork	[68]
red dry wine	4 days at 4 °C; vacuum	wild boars and deer	[78]
red wine	Step 1. 48 h at 4 °C Step 2. 7 days at 4 °C; without marinate	beef	[84]
red dry wine	7 days at 4 °C; vacuum	wild boars	[219]
red wine (dipping, immersing)	Step 1. Marination Variant 1. 15 min at 42 °C or 4 h at 4 °C; vacuum Variant 2. 4 °C during the entire experiment, or 15 min at 42 °C and storage at 4 °C. Step 2. Sonication: 25 kHz (300 W) and 1 MHz (150 W); 12 °C; 10 min	pork	[175]
unfiltered beers: India session ale and wheat ale (as base and oregano, parsley mustard, salt, pepper, garlic, olive oil, vinegar, and fresh onions)	12 h at 4 °C	moose and beef	[176,177,248,249,250]
pilsner beer, non-alcoholic pilsner, and black beer	4 h at 5 °C	pork	[251]
beer or red wine (as base and salt, garlic, sweet and hot paprika, sugar, coriander, mustard, marjoram, black pepper, juniper, onion, rosemary, clove, bay leaves, and monosodium glutamate)	12 h at 4 °C	pork	[252]
red wines (water or ethanol/water)	0.5, 3, and 24 h at 20 °C	chicken	[253]
red wine (as base and salt with various combinations of pomegranate ethanolic extract, or oregano and thyme essential oils)	24 h at 4 °C	pork	[254]
dry red wine (as base with lime tree honey, garlic, pepper, salt; with or without thyme, marjoram, or horseradish)	Step 1. 48 h at 4 °C Step 2. 3, 6, 9, and 12 days at 4 °C; vacuum	beef	[180] [255] [256]
red wine	0, 2, 4, 6, 8, and 10 days at 5 °C 0, 1, 2, 3, 4, and 5 days at 15 °C	beef	[257]
soy sauce, wine, high-fructose corn syrup, water, vinegar, salt, spices, onion powder, and garlic powder	7 days at 4 °C	beef	[258]

## Data Availability

Data are contained within the article. The data used to support the findings of this study can be made available by the corresponding author upon request.

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
