# Peer review of "Marinades Based on Natural Ingredients as a Way to Improve the Quality and Shelf Life of Meat: A Review"

_foods, 2023, doi:10.3390/foods12193638_

Round 1
Reviewer 1 Report
The subject of this paper, according to authors, was to provide a critical appraisal of meat marination process, including 65 different natural ingredients and the relationship between meat quality, its physicochemical and sensory indicators, and microbiological quality. The objective is really interesting and is focused on the preferable application of natural ingredients as effective preservatives through the marinades.
Nevertheless, there are significant issues, especially regarding the language and the multiple syntax and grammatical mistakes. The English should be carefully reviewed by an English native speaker (the MDPI tool might also be useful).
Examples of corrections to be performed: line 12, 28, 34, 49, 50, 59, 64, 70, 90-91, 139, lines 167-170 (rephrase, not clear), lines 201-202 (verb is missing), lines 301-303 (rephrase), 369, etc.
Another issue involves the organization and the structure of this review. Although providing abundant and interesting information, the way authors decided to present it, does not assist the reader to understand what type of marinade is ideal for each case, each type of product and especially, the type of deteriorative reaction leading to spoilage. In my opinion, general information as the one provided in lines 536-549, enriched with a brief presentation of the main spoilage mechanisms (microbiological decay, lipid oxidation, etc), depending on the type of meat, and the type of processing, eg grilling, thermal treatment, smoking is necessary in such a review. Such a separate section would enable to also show how different marinades can have a positive effect on product stability and shelf life.
Another point is that, in my opinion, in its present form, this is not actually a critical review, but mostly a report of recent findings in literature.
Fir example, authors could add a column in Tables 1-5, for each reference described, to summarize the main achievements obtained by using these marinades.
Furthermore, I believe that Figure 1 should be presented earlier in the text, and not as part of the Conclusion section. The Conclusion section should be re-visited as it is a repetition of the Introduction part.
In this section, it would be useful to critically summarize the results of the literature research, for example which type of marinade is mostly applied for a certain type of meat, how much can it prolong product shelf life, etc. What obstacles are there for an industrial application, especially when natural ingredients are used, instead of the typical preservatives, etc.
The subject of this paper, according to authors, was to provide a critical appraisal of meat marination process, including 65 different natural ingredients and the relationship between meat quality, its physicochemical and sensory indicators, and microbiological quality. The objective is really interesting and is focused on the preferable application of natural ingredients as effective preservatives through the marinades.
Nevertheless, there are significant issues, especially regarding the language and the multiple syntax and grammatical mistakes. The English should be carefully reviewed by an English native speaker (the MDPI tool might also be useful).
Examples of corrections to be performed: line 12, 28, 34, 49, 50, 59, 64, 70, 90-91, 139, lines 167-170 (rephrase, not clear), lines 201-202 (verb is missing), lines 301-303 (rephrase), 369, etc.
Another issue involves the organization and the structure of this review. Although providing abundant and interesting information, the way authors decided to present it, does not assist the reader to understand what type of marinade is ideal for each case, each type of product and especially, the type of deteriorative reaction leading to spoilage. In my opinion, general information as the one provided in lines 536-549, enriched with a brief presentation of the main spoilage mechanisms (microbiological decay, lipid oxidation, etc), depending on the type of meat, and the type of processing, eg grilling, thermal treatment, smoking is necessary in such a review. Such a separate section would enable to also show how different marinades can have a positive effect on product stability and shelf life. Another point is that, in my opinion, in its present form, this is not actually a critical review, but mostly a report of recent findings in literature. Fir example, authors could add a column in Tables 1-5, for each reference described, to summarize the main achievements obtained by using these marinades. Furthermore, I believe that Figure 1 should be presented earlier in the text, and not as part of the Conclusion section. The Conclusion section should be re-visited as it is a repetition of the Introduction part. In this section, it would be useful to critically summarize the results of the literature research, for example which type of marinade is mostly applied for a certain type of meat, how much can it prolong product shelf life, etc. What obstacles are there for an industrial application, especially when natural ingredients are used, instead of the typical preservatives, etc.
Author Response
Response to Reviewer 1
Thank you for reviewing our manuscript. Revisions were made according to the reviewers’ comments. Changes in the manuscript are indicated in Track revisions.
We hope that the improved manuscript will find your acceptance for publication. Thank you for your patience and help.
The subject of this paper, according to authors, was to provide a critical appraisal of meat marination process, including 65 different natural ingredients and the relationship between meat quality, its physicochemical and sensory indicators, and microbiological quality. The objective is really interesting and is focused on the preferable application of natural ingredients as effective preservatives throughthe marinades.Nevertheless, there are significant issues, especially regarding the language and the multiple syntax and grammatical mistakes. The Englishshould be carefully reviewed by an English native speaker (the MDPI tool might also be useful).
The text was proofread by Nicholas Paxford from Smaller Earth Poland.
Examples of corrections to be performed: line 12, 28, 34, 49, 50, 59, 64, 70, 90-91, 139, lines 167-170 (rephrase, not clear), lines 201-202 (verb is missing), lines 301-303 (rephrase), 369, etc.
Thank You for your comments. We corrected this
Another issue involves the organization and the structure of this review. Although providing abundant and interesting information, the way authors decided to present it, does not assist the reader to understand what type of marinade is ideal for each case, each type of product and especially, the type of deteriorative reaction leading to spoilage. In my opinion, general information as the one provided in lines 536-549, enriched with a brief presentation of the main spoilage mechanisms (microbiological decay, lipid oxidation, etc), depending on the type of meat, and the type of processing, eg grilling, thermal treatment, smoking is necessary in such a review. Such a separate section would enable to also show how different marinades can have a positive effect on product stability and shelf life.
Thank You for your comments. We agree with you, but our Review currently has 33 pages long, if we add new information, the manuscript wiil be too long. The cooking process is interesting, and the meat is not usually eaten raw, but maybe we will describe this in another manuscript.
Another point is that, in my opinion, in its present form, this is not actually a critical review, but mostly a report of recent findings in literature.
We have added new articles in the new version of the manuscript .
For example, authors could add a column in Tables 1-5, for each reference described, to summarize the main achievements obtained by using these marinades.
Thank You for your comments, but it is not easy, we tried to do it in previous version of the manuscript (before submission to the Journal), but in this case the tabels are unclear.
Furthermore, I believe that Figure 1 should be presented earlier in the text, and not as part of the Conclusion section.
We have moved Figure 1 (now Figure 2) to a new subsection
The Conclusion section should be re-visited as it is a repetition of the Introduction part.
We have shortened the conclusion as much as possible
In this section, it would be useful to critically summarize the results of the literature research, for example which type of marinade is mostly applied for a certain type of meat, how much can it prolong product shelf life, etc. What obstacles are there for an industrial application, especially when natural ingredients are used, instead of the typical preservatives, etc.
In our opinion, we have done this in the new version of the manuscript.
Reviewer 2 Report
This review manuscript by Latoch et al., provides a thorough overview of the using of natural ingredients of marinades, parameters of marinating process and mechanisms of changes in meat based on the most recently published scientific papers retrieved in the Web of Science Core Collection Database. The tables and the figure throughout the manuscript are very helpful, and the text does a good job of covering the vast array of marinating techniques and ingredients used in food industry and their effects on the meat structure. It is very helpful to have these approaches summarized in a single paper, and having a short overview of the mechanisms provides a better understanding of each approach. I support the possible further processing of the manuscript after appropriate modifications as outlined below:
1. The manuscript contains numerous errors in grammar, punctuation, and sentence structure. The author may seek assistance from a scientific writer or revise the manuscript themselves, ensuring all necessary corrections are made before resubmission.
2. L71: In the present form the “Materials and methods section” is unclear. The raw data processing after the first search results is unclear, specifically the eligibility criteria and rejection criteria of the cited works, number of chosen and rejected works. The articles have been selected with a solid and clear methodology? If yes, what? The “meat” term refers to the all food producing animals? The inclusion of an additional flowchart indicated the aforementioned issues would be useful.
3. L128: at the section “Fruit, vegetables marinade” I would like to suggest to the authors to consult and possible include other valuable recently published articles in the prestigious Foods journal in the reference list (e.g. doi: 10.3390/foods11030301)
3. L123: In the “3.2. Effect of marinades based on plants on quality of meat” section, please include other subheadings (e.g. L128- Fruit, vegetables marinade, L239:- Enzymes from fruit and vegetables, L274:- Enzymes from fruit and vegetables)
4. L126: the tables must be formatted according to the journal requirement
5. L316: for “164,165,166” or “192,193,194,195” the correct citations are “164-166” and “192-195”, please carefully revise this issue throughout the manuscript
Author Response
Response to Reviewer 2
Thank you for reviewing our manuscript. Revisions were made according to the reviewers’ comments. Changes in the manuscript are indicated in Track revisions.
We hope that the improved manuscript will find your acceptance for publication. Thank you for your patience and help.
The text was proofread by Nicholas Paxford from Smaller Earth Poland.
Comments and Suggestions for Authors
This review manuscript by Latoch et al., provides a thorough overview of the using of natural ingredients of marinades, parameters of marinating process and mechanisms of changes in meat based on the most recently published scientific papers retrieved in the Web of Science Core Collection Database. The tables and the figure throughout the manuscript are very helpful, and the text does a good job of covering the vast array of marinating techniques and ingredients used in food industry and their effects on the meat structure. It is very helpful to have these approaches summarized in a single paper, and having a short overview of the mechanisms provides a better understanding of each approach. I support the possible further processing of the manuscript after appropriate modifications as outlined below:
- The manuscript contains numerous errors in grammar, punctuation, and sentence structure. The author may seek assistance from a scientific writer or revise the manuscript themselves, ensuring all necessary corrections are made before resubmission.
The text was proofread by Nicholas Paxford from Smaller Earth Poland.
- L71: In the present form the “Materials and methods section” is unclear. The raw data processing after the first search results is unclear, specifically the eligibility criteria and rejection criteria of the cited works, number of chosen and rejected works. The articles have been selected with a solid and clear methodology? If yes, what? The “meat” term refers to the all food producing animals? The inclusion of an additional flowchart indicated the aforementioned issues would be useful.
We added a new text and Figure 1 entitled “Procedure for searching and selecting manuscripts for review”.
- L128: at the section “Fruit, vegetables marinade” I would like to suggest to the authors to consult and possible include other valuable recently published articles in the prestigious Foods journal in the reference list (e.g. doi: 10.3390/foods11030301).
We have added new articles marked in red color in the references section.
- L123: In the “3.2. Effect of marinades based on plants on quality of meat” section, please include other subheadings (e.g. L128- Fruit, vegetables marinade, L239:- Enzymes from fruit and vegetables, L274:- Enzymes from fruit and vegetables)
Thank you for your comments but we didn’t decide added additional subheadings.
- L126: the tables must be formatted according to the journal requirement
We corrected it.
- L316: for “164,165,166” or “192,193,194,195” the correct citations are “164-166” and “192-195”, please carefully revise this issue throughout the manuscript.
Thank you for your comments. We corrected it and checked the citations.
Reviewer 3 Report
This manuscript presents a comprehensive review of the marinade processing of meat. However, there are several points that authors need to address before the article be accepted for publication.
The authors stated that they critically appraised the meat marinating process. However, in the Abstract, only positive aspects are highlighted. Please add negative aspects and other reviewed data.
Please clarify what content is provided in Table 1. All or some collected literature data (Line 77).
Material and methods must be supplemented with exclusion criteria and an explanation of how data were summarized and presented within the Result section.
Authors should consider inserting the Table with traditional marinades, too.
Before the subsection 3.1. and next, the authors should briefly explain the parameters for evaluating meat quality.
Within subtitle 3.5, consider using “other fermented drink” instead of “natural products.”
The conclusion should be more concise.
Others:
Avoid starting sentences with the same word(s) within the same paragraph
Please check the guidelines for references.
The manuscript would benefit from being proofread by a native speaker to enhance the paper's readability.
Author Response
Response to Reviewer
Thank you for reviewing our manuscript. Revisions were made according to the reviewers’ comments. Changes in the manuscript are indicated in Track revisions.
We hope that the improved manuscript will find your acceptance for publication. Thank you for your patience and help.
Comments and Suggestions for Authors
This manuscript presents a comprehensive review of the marinade processing of meat. However, there are several points that authors need to address before the article be accepted for publication.
The authors stated that they critically appraised the meat marinating process. However, in the Abstract, only positive aspects are highlighted. Please add negative aspects and other reviewed data.
Thanks for your suggestions, we corrected it.
Please clarify what content is provided in Table 1. All or some collected literature data (Line 77).
Material and methods must be supplemented with exclusion criteria and an explanation of how data were summarized and presented within the Result section.
In the first version of the manuscript there was a table, we removed it, but in the text, in line 77, a reference to it remained. We added new information in section Material and methods according to Reviewer suggestions.
Authors should consider inserting the Table with traditional marinades, too.
Traditional marinades were not the subject of this manuscript. They were only shown as a background for natural marinades.
Before the subsection 3.1. and next, the authors should briefly explain the parameters for evaluating meat quality.
Thank you for your suggestions, but we decided not to add new text because the manuscript is too long.
Within subtitle 3.5, consider using “other fermented drink” instead of “natural products.” The conclusion should be more concise.
We agree with the Reviewer. The subtitle was changed in accordance with the Reviewer's suggestions.
Others: Avoid starting sentences with the same word(s) within the same paragraph
Thank you for your comments. We corrected it.
Please check the guidelines for references.
We checked the references and added new ones as suggested Reviewer.
Comments on the Quality of English Language -The manuscript would benefit from being proofread by a native speaker to enhance the paper's readability.
The text was proofread by Nicholas Paxford from Smaller Earth Poland.
Reviewer 4 Report
Authors are advised to arrange the review in following format, this will help to understand the past, current and future perspectives of this review. I can then give my opinion after changes
1. Introduction to Marinades and their Benefits
2. Natural Ingredients in Marinades: Exploring the Possibilities
3. Evaluating the Impact of Marinades on Meat Quality
4. Extending Shelf Life: The Role of Marinades
5. Popular Natural Ingredients for Marinades
6. Techniques for Marinating Meat with Natural Ingredients
7. Case Studies: Success Stories of Improved Meat Quality and Shelf Life
8. Considerations and Challenges in Using Natural Marinades
9. Future Directions and Research Opportunities
10. Conclusion: Harnessing the Power of Natural Ingredients in Marinades
Author Response
Response to Reviewer 4
Thank you for reviewing our manuscript. Revisions were made according to other reviewers’ comments. Changes in the manuscript are indicated in Track revisions.
We hope that the improved manuscript will find your acceptance for publication. Thank you for your patience and help.
Authors are advised to arrange the review in following format, this will help to understand the past, current and future perspectives of this review. I can then give my opinion after changes
- Introduction to Marinades and their Benefits
- Natural Ingredients in Marinades: Exploring the Possibilities
- Evaluating the Impact of Marinades on Meat Quality
- Extending Shelf Life: The Role of Marinades
- Popular Natural Ingredients for Marinades
- Techniques for Marinating Meat with Natural Ingredients
- Case Studies: Success Stories of Improved Meat Quality and Shelf Life
- Considerations and Challenges in Using Natural Marinades
- Future Directions and Research Opportunities
- Conclusion: Harnessing the Power of Natural Ingredients in Marinades
Thank you for your comments, we have revised our manuscript according to the comments of Reviewers 1,2 and 3, and we believe that our manuscript includes these points but in a different order.
Reviewer 5 Report
Regarding the manuscript entitled Marinades based on natural ingredients as a way to improve 2 the quality and shelf life of meat: A review. The review is interesting, but it is too long with many references. Please try to shorten it as possible. There are many unnecessary sentences. I recommend that the authors should double-check the language and grammar errors in their review.
The review is divided into introduction, materials and methods, results and conclusion. Where the discussion section? Change the heading to results and discussion.
L49. revise this sentence, language
L123. Plant-based marinades, correct
L128. Revise
I prefer figure 1 is not included in the conclusion please move it to appropriate place in previous sections
Conclusion section is so long and repetitive, please add strong recommendations and conclusion to the readers summarize what you have discussed in the review.
moderate English revision is required
Author Response
Response to Reviewer 5
Thank you for reviewing our manuscript. Revisions were made according to reviewers’ comments. Changes in the manuscript are indicated in Track revisions.
We hope that the improved manuscript will find your acceptance for publication. Thank you for your patience and help.
The text was proofread by Nicholas Paxford from Smaller Earth Poland.
Comments and Suggestions for Authors
Regarding the manuscript entitled Marinades based on natural ingredients as a way to improve 2 the quality and shelf life of meat: A review. The review is interesting, but it is too long with many references. Please try to shorten it as possible. There are many unnecessary sentences. I recommend that the authors should double-check the language and grammar errors in their review.
The review is divided into introduction, materials and methods, results and conclusion. Where the discussion section? Change the heading to results and discussion.
We corrected this sentence.
L49. revise this sentence, language We corrected this sentence.
L123. Plant-based marinades, correct We corrected it wherever necessary.
L128. Revise We have revised it.
I prefer figure 1 is not included in the conclusion please move it to appropriate place in previous sections.
We have moved Figure 2 to a new subsection
Conclusion section is so long and repetitive, please add strong recommendations and conclusion to the readers summarize what you have discussed in the review.
We have shortened the conclusion as much as possible
Comments on the Quality of English Language - moderate English revision is required
The text was proofread by Nicholas Paxford from Smaller Earth Poland.
Round 2
Reviewer 4 Report
In my opinion the authors made significant changes, and I agree to proceed for publication.
Author Response
Dear Reviewer,
Thank you very much for your positive reviews and help in improving our manuscript
Authors
Reviewer 5 Report
Dear authors
Thank you for your revisions
Minor editing
Author Response
Dear Reviewer
Thank you very much for your positive reviews and help in improving our manuscript. We have revised our manuscript with minor editing.
Authors